



# Anthropogenic CO₂-mediated freshwater acidification limits survival, calcification, metabolism, and behaviour in stress-tolerant freshwater crustaceans

Alex R. Quijada-Rodriguez[1], Pou-Long Kuan[2], Po-Hsuan Sung[2], Mao-Ting Hsu[2], Garett J.P. Allen[1], Pung Pung Hwang[3], Yung-Che Tseng[2*], Dirk Weihrauch[1*]

[1]Biological Sciences, University of Manitoba, Winnipeg, R3T 2N2, Canada
[2]Marine Research Station, Institute of Cellular and Organismal Biology, Academia Sinica, No. 23-10 Dawen Rd., Jiaoxi, Yilan County, Taiwan, 262
[3]Institute of Cellular and Organismal Biology, Academia Sinica, No. 128, Section 2, Academia Rd., Nangang District, Taipei City, Taiwan, 11529
* Indicates that these authors have contributed equally to this work

*Correspondence to*: Alex Quijada-Rodriguez (umquijaa@myumanitoba.ca, alexquijadarodriguez@gmail.com)

**Abstract.** Dissolution of anthropogenic CO₂ is chronically acidifying aquatic ecosystems. Studies indicate that ocean acidification will cause marine life, especially calcifying species, to suffer at the organismal and ecosystem levels. In comparison, freshwater acidification has received less attention rendering its consequences unclear. Here, juvenile Chinese mitten crabs, *Eriocheir sinensis*, were used as a calcifying model to investigate the impacts of CO₂-mediated freshwater acidification. Our integrative approach investigating changes in the animal's acid-base homeostasis, metabolism, calcification, locomotory behaviour, and survival rate indicate that the crab will face energetic consequences from future freshwater acidification. These energetic trade-offs allow the animal to maintain its acid-base homeostasis at the cost of reduced metabolic activity, exoskeletal calcification, and locomotion reducing the animal's overall fitness and increasing its mortality. Results suggest that present-day calcifying invertebrates could be heavily affected by freshwater acidification similar to their marine organisms and emphasizes the importance in understanding the long-term implications of freshwater acidification on species fitness.

## 1 Introduction

Rising levels of atmospheric CO₂ partially dissolve into marine systems causing a decrease in oceanic pH referred to as ocean acidification. In marine species, ocean acidification has been demonstrated to negatively impact development, metabolism, behaviour, and biomineralization potentially leading to major ecosystem-level changes (Kroeker et al., 2013; Melzner et al., 2009; Tresguerres and Hamilton, 2017). It is generally believed that freshwater systems will also experience acidification; however, the highly variable biogeochemistry between freshwater systems has been a limiting factor in modelling future freshwater scenarios (Hasler et al., 2016; Phillips et al., 2015; Weiss et al., 2018). Two recent case studies on different freshwater systems have suggested that the magnitude of CO₂ mediated acidification could be similar or even exceed predicted



levels of ocean acidification (Phillips et al., 2015; Weiss et al., 2018). The potential that freshwater acidification may be of equal or greater severity than ocean acidification emphasizes the need to understand the biological responses and consequences to freshwater species.

Amongst marine organisms, calcifying species are particularly sensitive to acidification as dissolution of $CO_2$ reduces carbonate availability in parallel to pH, potentially increasing dissolution of their calcified exoskeleton (Feely et al., 2004; Roleda et al., 2012). To date there are no comprehensive studies investigating the various physiological and behavioural effects of realistic future levels of $CO_2$-mediated acidification in calcifying freshwater invertebrates. Freshwater calcifying macroorganisms are largely limited to crustaceans and molluscs that comprise roughly 10% and 4% of freshwater species

diversity, respectively (Balian et al., 2008). Crustaceans are arguably one of the most successful animal groups having occupied almost all ecological niches across the globe including freshwater, marine, and terrestrial habitats making them a good model to study global change consequences in physiologically and ecologically robust group of species. Freshwater crustaceans occupy a key position in food webs where all crustacean life stages provide a vital food source for a wide range of juvenile and adult predators (Cumberlidge et al., 2009). Additionally, freshwater crustaceans provide vital ecological services as

indicators of water quality, nutrient cycling of detritus and bioturbation of sediment (Cumberlidge et al., 2009). From an economic standpoint, freshwater crustaceans account for ~30% (2.5 million tons) of aqua-cultured crustaceans worldwide demonstrating that this group is an important human food source (Tacon, 2020). The ecological and economic importance of freshwater crustaceans together with the general sensitivity of calcifying species to acidification makes it imperative to determine how freshwater crustaceans respond to anthropogenic $CO_2$-mediated acidification of their environment as this may

provide insights to understand how global change may affect freshwater ecosystem dynamics.

Here we investigated the effects of a potential future $CO_2$-mediated freshwater acidification scenario on the acid-base regulation, metabolism, calcification, behaviour, and survival rate in the juvenile life stage of the highly invasive catadromous Chinese mitten crab (*Eriocheir sinensis*). Native to China's Yangtze river system, the third-largest river system in the world, juvenile Chinese mitten crab in this habitat already experience regular fluctuations in freshwater $pCO_2$ from 681-3796μatm

(Ran et al., 2017), which may confer some degree of pre-adaptation to elevated $CO_2$ due to life history. Furthermore, crustaceans are generally believed to more $CO_2$ tolerant than other calcifying organisms such as bivalves and coral due to their high metabolic activity and robust acid-base machinery allowing for a more efficient compensation of acid-base disturbances (Melzner et al., 2009). These combined predictors of $CO_2$ tolerance make Chinese mitten crab an interesting model to study the effects of future $CO_2$ mediated freshwater acidification as they may already possess the adaptations necessary to deal with

future freshwater acidification conditions. Therefore, we hypothesized that freshwater crustaceans such as Chinese mitten crab would be well-adapted to counteracting challenges associated with fluctuating $pCO_2$ resulting from anthropogenic activity and not experience detrimental physiological or behavioural impairment.



## 2 Methods

### 2.1 Animal Maintenance

Male and female juvenile Chinese mitten crab (*Eriocheir sinensis* 10-20g) were purchased from the Chinese mitten crab Breeding Association of Taiwan. Crabs were maintained at the Academia Sinica Institute of Cellular and Organismal Biology aquatics facility (Taipei, Taiwan) in three 120-L aquariums with flow through dechlorinated Taipei tap water (henceforth referred to as freshwater) at $23 \pm 0.5°C$ on a 14:10h light-dark cycle. Water parameters for these holding tanks were the same as that of the control water used in the experimental acclimations. Juveniles crabs in non-experimental holding tanks were

maintained at a density of roughly 100 individuals per tank with a constant flow of freshwater to prevent the build up of metabolic wastes. Crabs were fed *ad libitum* with oatmeal and mollusc meat three times per week and monitored for activity level and the presence of disease as general health indicators. Crabs were fasted for a minimum of 48 hours prior to sampling to minimize the effects of dietary intake on measured parameters.

### 2.2 Freshwater acidification

For experimental acclimation, crabs were sampled upon removal from the holding tanks (0-day time point) and transferred to flow through 10-L experimental tanks (6-7 crabs per tank, 4 tanks per treatment) containing either control or acidified freshwater (Table 1). Acidified freshwater was achieved by injection of $CO_2$ directly into the experimental tanks by air stone to maintain a pH of 6.73 (pH controller, MACRO). $CO_2$ bubbling rate and freshwater flow rate was adjusted to minimize overshooting the target $pCO_2$ level. Following injection of $CO_2$ to regulate water $pCO_2$ we recorded a brief $pCO_2$ overshoot to

a maximum level of 570Pa resulting from direct $CO_2$ injection into the experimental tanks by the pH controller. Water pH, total alkalinity, and temperature were measured daily in the experimental tanks. Water pH (NBS scale) and temperature were measured with a pH electrode (Accumet AP55 pH/ATC electrode, Ohio, USA) connected to a portable pH meter (Accumet AP71, Ohio, USA) calibrated with pH buffers (pH 4.00, 7.00, and 10.01) traceable to NIST standard reference material (Thermofisher Orion). Water alkalinity was measured by spectrophotometric assay on a Nanodrop 2000c (Thermoscientific,

Wilminton, DE, USA) according to previously established protocols (Sarazin et al., 1999). Water $pCO_2$ was calculated with the CO2SYS excel add-in (Lewis and Wallace, 1998) using measured water temperature, pH and total alkalinity. Constants used for $pCO_2$ calculations include freshwater carbonate dissociation constants ($K_1$ and $K_2$) from Millero (1979), and $KHSO_4$ constants from Dickson (1990).

**Table 1. Measured tank parameters for control and $CO_2$ acidified freshwater (FW).**

|  | Temp (˚C) | pH | TA (µmol l⁻¹) | TCO₂ (µmol l⁻¹) | pCO₂ (µatm) |
|---|---|---|---|---|---|
| Control FW | $23 \pm 0.5$ | $7.41 \pm 0.02$ | $501.3 \pm 32.0$ | $547.2 \pm 35.8$ | $1301 \pm 122$ |
| Acidified FW | $23 \pm 0.5$ | $6.74 \pm 0.01$ | $423.2 \pm 11.1$ | $602.5 \pm 2.3$ | $4994 \pm 114$ |






## 2.3 Hemolymph Acid-Base Status

Hemolymph samples (100µL per crab) were taken at the base of a walking leg with a sterile syringe according to previous protocols for *E. sinensis* (Truchot, 1992). Samples from 2-3 crabs were pooled together (200-300uL pooled hemolymph per n value) to obtain enough sample for downstream analysis of ammonia, pH and total carbon. Pooled hemolymph samples were

gently mixed by slowly pipetting to avoid off gassing of $CO_2$ and thereby disrupting hemolymph acid-base parameters. Measurements of pH and total carbon were performed immediately after hemolymph collection and the remaining hemolymph was frozen -20°C for later analysis of ammonia. Hemolymph pH (200-300uL samples) was measured in NBS scale using an InLab micro pH electrode calibrated with pH buffers traceable to NIST standard reference material (Thermofisher Orion). Hemolymph total carbon was measured in duplicate (50uL per measurement) using the Corning 965 carbon dioxide analyser

(±0.2mM precision) calibrated with $NaHCO_3$ standards ranging from 0-20mmol $l^{-1}$ to produce a standard curve with a minimum $R^2$ of 0.99. Hemolymph $pCO_2$ and $HCO_3^-$ were calculated using a rearrangement of the Henderson-Hasselbalch equation with pK1 and $\alpha CO_2$ values derived for *E. sinensis* hemolymph at 23°C (pK1= 6.079773, $\alpha CO_2$= 0.00031263mmol $l^{-1}$ $Pa^{-1}$ (Truchot, 1976, 1992). Hemolymph ammonium was measured in triplicate (25uL hemolymph per measurement) with a microplate reader (Molecular Devices, SpectraMax, M5) using an orthophthaldialdehyde fluorometric assay which is

insensitive to amino acids and proteins (Holmes et al., 1999). Ammonia standards were made from $NH_4Cl$ in *E. sinensis* ringer (pH 8.1) containing (in mmol $l^{-1}$): 185 NaCl, 16 $CaCl_2$, 6 $MgCl_2$, 7 KCl, and 13 $NaHCO_3$. The ion concentrations for the ringer were based on ion composition measurement done on 4 juvenile Chinese mitten crab in this study (in mmol $l^{-1}$ $Na^+$ 191, $K^+$ 7.2, $Ca^{2+}$ 16.3, $Mg^{2+}$ 5.9, $Cl^-$ 252). Concentrations of $Na^+$, $K^+$, $Ca^{2+}$, $Mg^{2+}$ were measured by flame absorption spectrophotometry (Polarized Zeeman Atomic Absorption Spectrophotometer Z-5000, Hitachi High-Technologies, Tokyo,

Japan), $Cl^-$ was measured spectrophotometrically using the mercury (II) thiocyanate method (Florence and Farrar, 1971). $HCO_3^-$ and pH values for the ringer were based on measurements taken from control crabs at 0 days in this study and measured as described above.

## 2.4 Ammonia excretion and metabolic rate

Ammonia excretion and standard metabolic rate (SMR) were measured over the time course of a one-week acclimation to

control and acidified freshwater. These two parameters were measured on individual crabs randomly selected from the four control and four acidified freshwater aquaria. Experimental sampling of ammonia excretion and SMR were performed in parallel to hemolymph sampling however, crabs were first randomly selected and placed into respirometry chambers before selecting crabs for hemolymph sampling to avoid using crabs recently sampled for hemolymph. Ammonia excretion experiments were performed in plastic Tupperware completely filled with 200mL filtered control or acidified freshwater. Crabs

were given 30 minutes to acclimate to the experimental chambers prior to initiation of water sampling as ammonia excretion is elevated for a short time directly after handling (Hans et al., 2014). Water samples (1mL) for ammonia analysis were collected directly after 30 and 90 minutes of being placed in the experimental chambers. Ammonia concentrations of the water





at the 30 and 90 minute time points were determined using the aforementioned orthophthaldialdehyde fluorometric assay (Holmes et al., 1999). Ammonia excretion rates were calculated according to the Eq. (1):

$Ammonia\ excretion\ rate = \frac{([Amm_{90}] - [Amm_{30}]) * V}{t * m},$  (1)

where $Amm_{90}$ is the water ammonia concentration at 90 minutes, $Amm_{30}$ is the water ammonia concentration at 30 minutes, V is the chamber volume during the flux period in litres, t is the flux time in hours, and m is the fresh weight of the crab in grams.

Standard metabolic rate was measured by closed system respirometry in custom 3L glass respiration chambers containing
0.2µm filtered freshwater control or acidified freshwater. Crabs were given 15 minutes to adjust to fully oxygenated respiration chambers before sealing chambers while submerged in a bath of control or acidified freshwater. Chambers were placed horizontally allowing for lateral crab movement in the chamber and oxygen saturation was continuously measured every 15 seconds for 30 minutes at 23°C. The oxygen sensor (PreSens oxygen micro optode, type PSt1, PreSens Precision Sensing GmbH, Regensburg, Germany) was attached to the top of the chamber and connected to an OXY-4 mini multichannel fiber
optic oxygen transmitter (PreSens Precision Sensing GmbH, Regensburg, Germany). Oxygen saturation was never allowed to drop below 80% and respiration chambers without a crab were used to determine any potential background bacterial respiration for each trial. Preliminary trials demonstrated that crab movement and ventilation rate in the chamber was sufficient to mix the water within the chamber and prevent oxygen stratification as indicated by a linear decline in oxygen availability.

**2.5 Carapace calcification**

To assess carapace calcification, changes in the calcium content relative to carapace mass was measured according to previously established protocols (Spicer and Eriksson, 2003). In brief, a piece of carapace (ca. 2.5cm$^2$, 15.2 ± 0.4mg) was removed from the dorsal carapace. The weighed piece of carapace was digested in HNO$_3$ (13.1 N) at 60°C for 16 hours. Digested samples were then diluted to a final HNO$_3$ concentration of 2% v/v. The carapace Ca$^{2+}$ content was measured by atomic absorption spectrophotometer (Z-8000; Hitachi). Standard solutions from Merck (Darmstadt, Germany) were used to
make the Ca$^{2+}$ standard curve.

**2.6 Locomotory Behaviour Assay**

A 24 x 24 cm square, novel, opaque tank was used in the open field test to assess changes in movement of juvenile crabs exposed to control and freshwater acidified conditions for one week. Acclimated crabs were transferred to the novel tank containing control or acidified freshwater and given 5 minutes to acclimate as done in previous crustacean behavioural studies
(Robertson et al., 2018). After acclimation, crab activity was recorded with a digital camera (UI-3240CP Rev.2, Ids, Germany) for 5 minutes (300 seconds) and videos of the movement were processed with the image analysis Ethovision XT motion tracking software (v. 7.0, Noldus, Netherlands). In this study 4 factors were measured; distance covered (cm), velocity (cm/s), movement (time in movement, seconds) and mobility (time in mobile state, seconds). We defined movement as the duration





for which the central body point (whole body) was changing location. Mobile state was defined as the duration in which crabs

exhibited any movement even if the center point of the animals remained in the same location for example, appendage movement.

## 2.7 Statistical analysis

Statistical analyses were conducted using JMP Pro 15 (Cary, NC, USA) and GraphPad Prism 8.4.2 (San Diego, CA, USA). Data were analysed for outliers by ROUT test with a Q value of 1%. For all data heterogenicity of variance was tested by

Levene's test and normal distribution of residuals by Shapiro-Wilk test. For this study, time course data was analysed by one-way ANOVA for freshwater acidified crabs and control crabs independently as opposed to a two-way ANOVA as the control crabs measurements were done solely to demonstrate that measured parameters do not change over time and not for a cross comparison between the two groups. The exception to this was the calcification data which was analysed by two-way ANOVA as a 0-week point was not measured thereby requiring comparison between the two crab groups. Data for hemolymph pH,

$HCO_3^-$, ammonia and SMR satisfied the assumptions for parametric analysis and were analysed by one-way ANOVA post hoc Dunnett's test with pairwise comparisons made to the 0-day time point. Data for hemolymph $pCO_2$ and ammonia excretion rates did not meet the assumptions for parametric analysis so were analysed by Kruskal-Wallis test post hoc Steel test with pairwise comparisons made to the 0-day time point. Carapace calcification data was analysed by two-way ANOVA post hoc Tukey HSD with time and $pCO_2$ level used as the fixed factors. Behavioural data was analysed by student's t-test with the

exception of appendage movement time that did not meet parametric assumptions so was analysed by Wilcoxon test. Survival curves were analysed for significant differences by the Mantel-Cox test and hazard ratio was determine by the Mantel-Haenszel test. For all data sets, $p$ values $\leq 0.05$ were considered significant. Data are presented as mean ± standard error (SEM). Statistical output results are written in text or summarized in Table 2.

## 3 Results

### 3.1 Probability of Survival

The effect of freshwater acidification on survival was determined by generating survival curves for crabs in control and acidified freshwater (Fig. 1). There was a significant difference in the probability of survival between the control and acidified freshwater environments (Mantel-Cox log rank test, $X^2_1$=9.41, p=0.0022, Fig. 1), with a 50% mortality in crabs held in the acidified freshwater compared to 15% mortality in control freshwater. Calculation of the Mantel-Haenszel hazard ratio

indicates that crabs in acidified freshwater have a 3.68 times greater probability of mortality than the crabs held under control conditions.

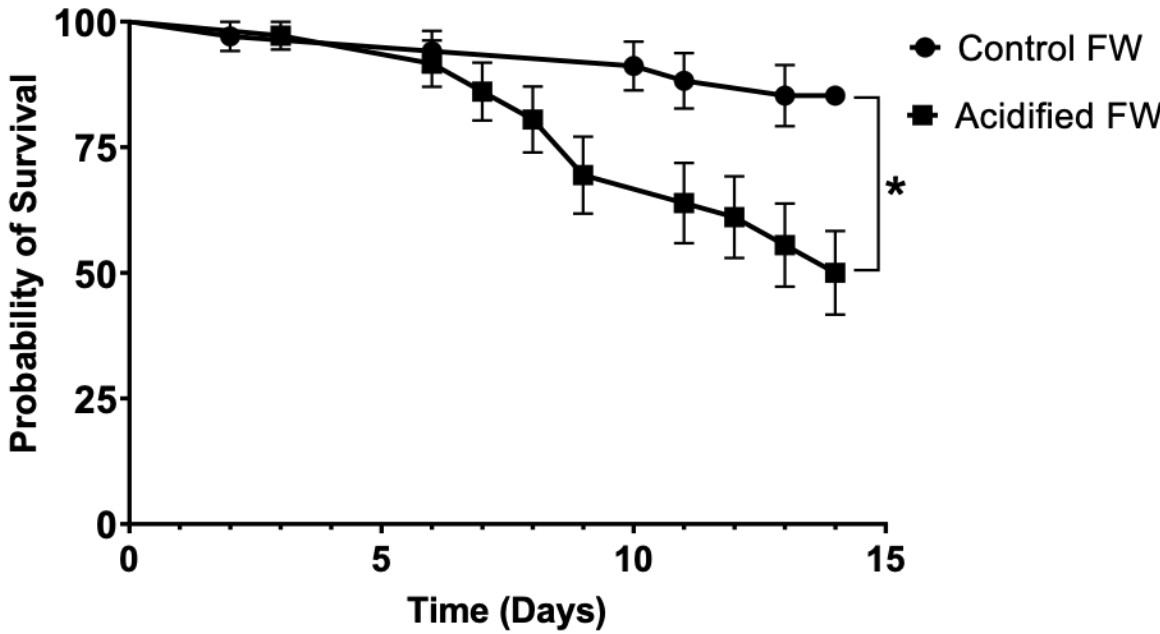

**Figure 1. Survivorship curves of juvenile Chinese mitten crab, *Eriocheir sinensis*, over 14 days of exposure to control (pH 7.41, 1301μatm pCO$_2$) or CO$_2$-acidified (pH 6.74, 4994μatm pCO$_2$) freshwater. Data are presented as probability of survival +/- SE. (N=34 for control freshwater and N=36 for acidified freshwater). Statistical significance was assessed by Mantel-Cox test * indicating significant difference between probability of survival between control and freshwater acidified crab populations.**

### 3.2 Acid-base status

Chinese mitten crab maintained in control freshwater showed no changes in hemolymph pH, bicarbonate, pCO$_2$, or ammonia throughout the experimental time course (Fig. 2; Table 2). In contrast, acidified freshwater had a significant effect on hemolymph pH, bicarbonate, pCO$_2$, or ammonia (Fig. 2; Table 2). Exposure to acidified freshwater induced a respiratory acidosis indicated by a decline in hemolymph pH (pH 8.11 ± 0.015 to 8.03 ± 0.0019) and a transient rise in hemolymph pCO$_2$ (404 ± 23Pa to 486 ± 26Pa) within the first six hours of exposure (Fig. 2a, c). This acidosis was maintained for two days with full recovery occurring by day seven of exposure when hemolymph pH returned to control levels, although hemolymph pCO$_2$ remained elevated (499 ± 20Pa). Recovery of hemolymph pH coincided with increases in hemolymph HCO$_3^-$ (16.7 ± 0.78mmol l$^{-1}$) and ammonia (136 ± 2.9μmol l$^{-1}$; Fig. 2b,d); however, no significant changes in hemolymph HCO$_3^-$ and ammonia were observed during the first two days of exposure suggesting a delayed extracellular pH regulatory response.





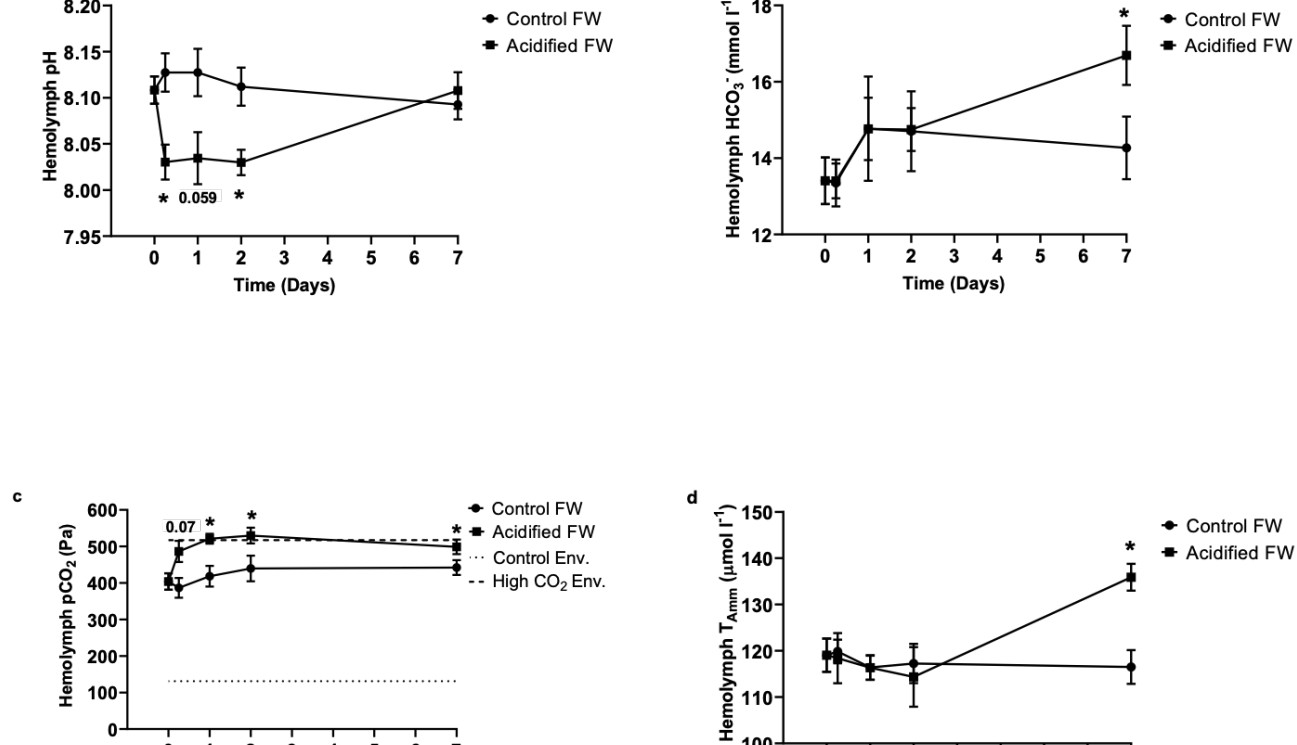

**Figure 2. Changes in extracellular (a) pH, (b) $HCO_3^-$, (c) $pCO_2$, and (d) ammonia of juvenile Chinese mitten crab, *Eriocheir sinensis*, during a 7-day time course of exposure to control (pH 7.41, 1301μatm $pCO_2$) or $CO_2$-acidified (pH 6.74, 4994μatm $pCO_2$) freshwater. Data are presented as mean +/- SEM. (N=6-14, 2-3 crabs pooled per N value). Statistical significance was assessed by one-way ANOVA followed by a post-hoc Dunnett's test with * indicating significant difference from day 0 measurements. P-values near but not <0.05 are written above corresponding data point.**

## 3.3 Metabolism

Metabolic changes were quantified through an individuals' ammonia excretion rate as an indicator of potential shifts in protein catabolism and their oxygen consumption rate as an indicator of changes in aerobic metabolism. Control crabs exhibited steady oxygen consumption rates and ammonia excretion rates throughout the measured time course (Fig. 3; Table 2). In contrast, crabs exposed to freshwater acidification experienced a significant reduction in oxygen consumption rate within six hours that was maintained throughout the remainder of the time course (Fig. 3a; Table 2). Ammonia excretion rates were significantly affected by acidified freshwater (Fig. 3b; Table 2.). Initially excretion rates were unchanged until a transient increase in the second day of freshwater acidification exposure when excretion rates nearly doubled and were significantly higher than control levels by day seven (Fig. 3b).





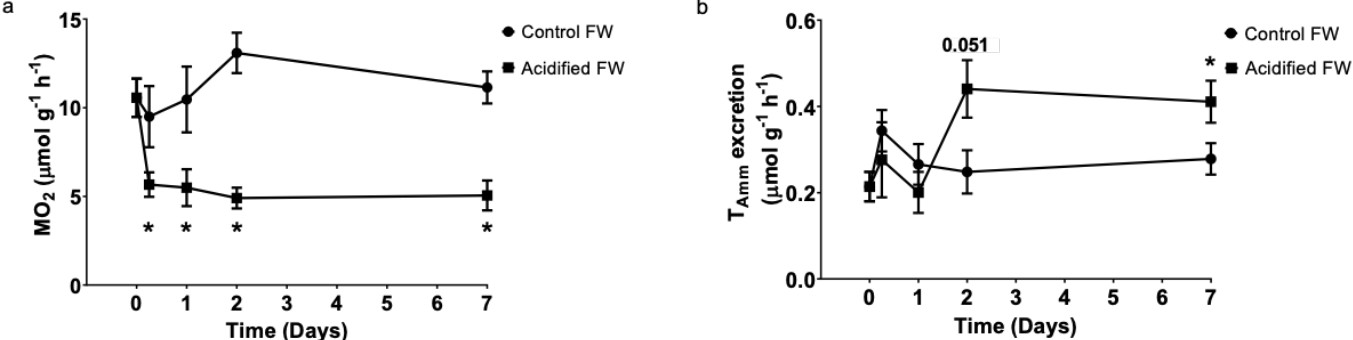

**Figure 3. Changes in whole animal (a) oxygen consumption rate (MO₂) and (b) ammonia excretion rate of juvenile Chinese mitten crab, *Eriocheir sinensis*, during a 7-day time course of exposure to control (pH 7.41, 1301μatm pCO₂) or CO₂-acidified (pH 6.74, 4994μatm pCO₂) freshwater. Data are presented as mean +/- SEM. (N=5-6 for oxygen consumption and N=7-12 for ammonia excretion). Statistical significance was assessed by one-way ANOVA followed by a post-hoc Dunnett's test for MO2 and Kruskal-Wallis post hoc Steel test for ammonia excretion rates. Significant differences from day 0 measurements are indicated by \*. P-values**
**near but not <0.05 are written above corresponding data point.**

### 3.4 Carapace calcification

Changes in calcification were quantified as the change in the crab's exoskeletal calcium content following exposure to freshwater acidification conditions. Calcification was measured several times over a six-week acclimation as several studies

on marine crustaceans report changes in calcification after 20+ days of acclimation (Long et al., 2013; Ries et al., 2009; Taylor et al., 2015). Overall, there was a significant time, pCO₂ and interactive time and pCO₂ effect on calcification (Table 2). Post hoc analysis suggests there were no significant changes in carapace calcification in the first two weeks of exposure to freshwater acidification (Fig. 4). However, after three and six weeks of exposure, a significant decline in carapace calcium content to 84.1 ± 2.9% and 85.2 ± 3.3% of control crab levels was observed (Fig. 4).

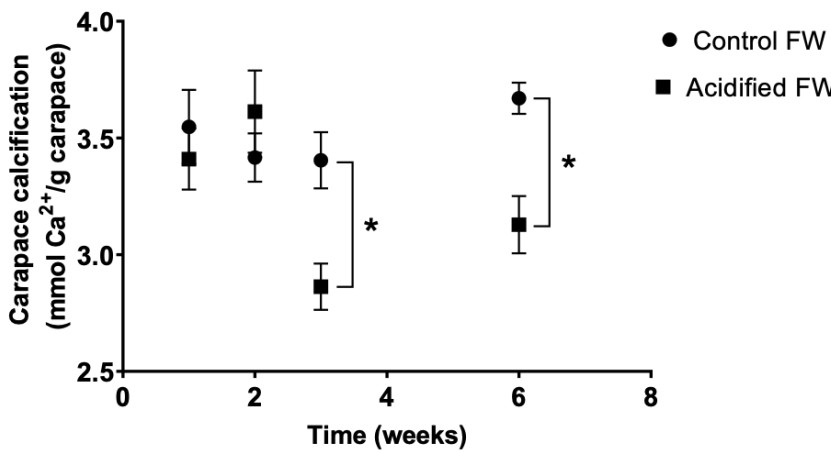






**Figure 4. Changes in carapace calcium content of juvenile Chinese mitten crab, *Eriocheir sinensis*, over a 6-week exposure to control (pH 7.41, 1301μatm $pCO_2$) or $CO_2$-acidified (pH 6.74, 4994μatm $pCO_2$) freshwater. Data are presented as mean +/- SEM. (N=6-12). Statistical significance was assessed by two-way ANOVA followed by a post-hoc Tukey HSD test with * indicating significant difference between control and acidified FW crabs for each respective week.**


**Table 2. Results of ANOVA and Kruskal-Wallis statistical analyses performed in the study. P-values below 0.05 are considered statistically significant and are bolded.**

One-way ANOVA

| Response Variable | Treatment | df | $df_{error}$ | F ratio | p-value |
|---|---|---|---|---|---|
| Hemolymph pH | Control FW | 4 | 40 | 0.45 | 0.77 |
| | Acidified FW | 4 | 44 | 4.91 | **0.0023** |
| Hemolymph $HCO_3^-$ | Control FW | 4 | 40 | 0.64 | 0.64 |
| | Acidified FW | 4 | 44 | 4.85 | **0.0025** |
| Hemolymph Ammonia | Control FW | 4 | 36 | 0.18 | 0.95 |
| | Acidified FW | 4 | 38 | 5.56 | **0.0013** |
| SMR | Control FW | 4 | 23 | 0.99 | 0.43 |
| | Acidified FW | 4 | 24 | 6.96 | **0.0007** |

Kruskal-Wallis Test

| Response Variable | Treatment | df | $X^2$ | p-value | |
|---|---|---|---|---|---|
| Hemolymph $pCO_2$ | Control FW | 4 | 3.06 | 0.55 | |
| | Acidified FW | 4 | 13.59 | **0.0087** | |
| Ammonia excretion rate | Control FW | 4 | 5.14 | 0.27 | |
| | Acidified FW | 4 | 14.13 | **0.0069** | |

Two-Way AVOVA

| Response Variable | Independent variable | df | $df_{error}$ | F ratio | p-value |
|---|---|---|---|---|---|
| Carapace $Ca^{2+}$ | Time | 3 | 68 | 3.98 | **0.011** |
| | $CO_2$ | 1 | 68 | 8.59 | **0.0046** |
| | Time x $CO_2$ | 3 | 68 | 4.33 | **0.0074** |

## 3.5 Locomotory Behaviour Assay

An open field test was used to quantify locomotory behavioural changes over a five-minute recording period in a novel arena

(Table 3). Crabs exposed to acidified freshwater on average moved less distance in the novel arena than crabs in control freshwater (student's t-test, $t_{35}$=-2.5, p=0.017, Table 3). Crabs in acidified freshwater also had a lower velocity than crabs in





control freshwater after the seven-day exposure (student's t-test, $t_{35}$=-2.37, p=0.024, Table 3). Movement and mobility were also quantified, where movement was defined as the crab changing its relative location in the arena and mobility was defined as the movement of body appendages even if the crab's location did not change. There was a significant decrease in movement

(student's t-test, $t_{35}$=-2.55, p=0.015, Table 3) and mobility (Wilcoxon test, Z=2.08, p=0.037, Table 3) following the seven-day exposure acidified freshwater.

**Table 3. Changes in locomotory behaviour of juvenile Chinese mitten crab, Eriocheir sinensis, after a seven-day exposure to control (pH 7.41, 1301µatm $pCO_2$) or $CO_2$-acidified (pH 6.74, 4994µatm $pCO_2$) freshwater. Data are presented as mean +/- SEM. (N=18-19). Statistical significance was assessed by student's t-test or Wilcoxon test for mobility time with * indicating significant difference**
**between control and acidified FW treatments.**

| | Distance moved (cm) | Velocity (cm $s^{-1}$) | Movement time (s) | Mobility time (s) |
|---|---|---|---|---|
| Control FW | $761 \pm 46$ | $2.53 \pm 0.15$ | $148 \pm 7$ | $215 \pm 5$ |
| Acidified FW | $601 \pm 45*$ | $2.04 \pm 0.15*$ | $119 \pm 9*$ | $179 \pm 14*$ |

## 4 Discussion

Anthropogenically driven aquatic acidification has the potential to negatively impact both freshwater and marine life. Meta-analyses of biological responses to ocean acidification suggest that marine crustaceans generally experience minimal consequences to $pCO_2$ tensions (~1000µatm) predicted to occur by the year 2100 with further acidification to levels expected
for year 2300 (~2000µatm) negatively impacting about half of the studied marine crustaceans (Kroeker et al., 2013; Melzner et al., 2009; Wittmann and Pörtner, 2013). In contrast, the biological responses of any freshwater invertebrate to realistic future $CO_2$ mediated freshwater acidification remains unknown. In the present study, we aimed to demonstrate for the first time the physiological and behavioural consequences of a possible future $CO_2$ mediated freshwater acidification scenario on a juvenile calcifying invertebrate, the Chinese mitten crab, *Eriocheir sinensis*. Our results suggest that freshwater juvenile Chinese mitten
crab experience significant impairment of metabolism, calcification, locomotory behaviour and survival when exposed to a potential future freshwater acidification (4994µatm $pCO_2$). While extracellular acid-base status is successfully regulated, the high energetic demands to sustain essential physiological processes such as acid-base regulation may be causing energetic reallocation that leads to a trade-off that impairs several physiological processes and alters animal fitness.

### 4.1 Plausibility of Freshwater Acidification Conditions

Modelling of future $CO_2$ mediated freshwater acidification for the year 2100 is nearly non-existent making the plausibility of the $pCO_2$ levels used in this study difficult to assess. The control $pCO_2$ levels used in this study reflect the average $pCO_2$ measured in 13 stations along the mainstem of the Yangtze River system (excluding Nanjing station which is at the mouth of the river and influenced by coastal upwelling) (Ran et al., 2017). The future freshwater acidification conditions used in this



study represents a 3691µatm increase in $pCO_2$ from control levels and roughly 1200µatm higher than the highest average

level recorded by the 13 stations along the mainstem of the Yangtze river (Ran et al., 2017). While future $CO_2$ mediated acidification models are not available for the Yangtze river, the relationship between changes in freshwater $pCO_2$ in other freshwater systems as a response to changes in atmospheric $pCO_2$ may provide indications of plausible future increases in $pCO_2$. Weiss et al. (2018) tracked changes in $pCO_2$ of four freshwater bodies in Germany between 1981-2015 and reported that freshwater $pCO_2$ increased by an average of 561µatm over this time period while atmospheric $pCO_2$ increased by ~60µatm

from 340 to 399.42µatm (National Oceanic and Atmospheric Administration; www.esrl.noaa.gov/gmd/dv/iadv). This relationship suggests that for every 1µatm increase in atmospheric $pCO_2$, these freshwater bodies increased by 9.35µatm. Since atmospheric $pCO_2$ is projected to rise to approximately 985µatm by the year 2100 (IPCC, 2013) this would mean that freshwater $pCO_2$ in these systems could rise by as much as 5469µatm. Assuming this relationship is accurate, the $pCO_2$ levels used in this study would be within a range that could feasibly occur in the Chinese mitten crab's native environment by the

year 2100. Further, it should be noted that while freshwater systems average $pCO_2$ levels of 3100µatm (streams and rivers) and 1410µatm (lakes), the $pCO_2$ levels used for acidified freshwater in this study are within ranges that can already be seen in freshwater systems globally for example the Mackenzie, Mississippi, Ohio and Elbe rivers suggesting that acidification scenario (Cole and Caraco, 2001; Raymond et al., 2013).

**4.2 Probability of Survival**

Sensitivity to aquatic acidification is quite variable in marine crustaceans. In mid to high intertidal and burrowing species including porcelain crabs (*Petrolisthes cinctipes*, *Petrolisthes manimaculus*, and *Porcellana platycheles*), burrowing shrimp (*Upogebia deltaura*), and barnacles (*Semibalanus balanoides* and *Elminius modestus*), minimal changes in survival probability are reported at $pCO_2$ tensions ranging from 1395-2707µatm (Donohue et al., 2012; Findlay et al., 2010; Page et al., 2017). Presumably the variability in $CO_2$ levels experienced in burrows and intertidal zones has driven the evolution of adaptation for

greater $CO_2$ tolerance in these groups of crustaceans. We predicted that juvenile Chinese mitten crab would also have an elevated $CO_2$ tolerance and face minimal changes in survival probability due to freshwater acidification as the natural habitat of these crustaceans, the Yangtze river, is known to fluctuate by as much as 3000µatm (Ran et al., 2017). Despite being a freshwater organism with strong ionoregulatory capabilities to deal with environmental acidification our results alarmingly show a sharp decrease in survival rate of Chinese mitten crabs over a 14-day period of exposure to 4994µatm $pCO_2$ (Fig 1).

Such rapid decreases in survival have also been observed in non-burrowing crustaceans or crustaceans that do not inhabit high intertidal regions including brine shrimp (*Artemia sinica*), red king crab (*Paralithodes camtschaticus*), and low intertidal long-clawed porcelain crab, (*Pisidia longicornis*), exposed to 1500, 1637, and 5821µatm $pCO_2$, respectively (Long et al., 2013; Page et al., 2017; Zheng et al., 2015). It might be tempting to conclude that low survival in Chinese mitten crabs compared to tolerant mid to high intertidal and burrowing marine crustaceans is simply due to the greater $pCO_2$ tensions used in the present

study (4994µatm), however, even higher levels (5821µatm) have been shown to have no effect on the probability of survival in a tolerant intertidal broad-clawed porcelain crab (*Porcellana platycheles*) after a 24-day exposure (Page et al., 2017).





Therefore, the low survival rates in the present study suggest a high susceptibility to acidification and are not consistent with the hypothesis that inhabiting a highly fluctuating $CO_2$ environment confers tolerance to future freshwater acidification.

### 4.3 Physiological Responses

Juvenile Chinese mitten crab effectively recovered extracellular pH following respiratory acidosis resulting from freshwater acidification by accumulation of extracellular $HCO_3^-$ as a buffer (Fig. 2). Compensation of acid-base homeostasis under freshwater acidification was not surprising given that strong acid-base regulatory capabilities are typically seen in highly active organisms such as fish, cephalopods and crustaceans (Melzner et al., 2009). Similar recovery of extracellular pH to elevated environmental $CO_2$ has also been observed in Dungeness crab (*Metacarcinus magiste*r) and velvet crab (*Necora pub*er) exposed

to even higher $pCO_2$ tensions (10000+µatm; Pane and Barry, 2007; Spicer et al., 2007). In contrast, green crab (*Carcinus maenas*) and blue crab (*Callinectes sapidus*) have been shown to not fully compensate extracellular pH at 10000+µatm $CO_2$ levels (Cameron, 1978; Fehsenfeld and Weihrauch, 2016); however, measurement in these species were only done over 48 hours and more time may have been required for the animals to recover as seen in our study where recovery was only observed after seven days. The compensatory responses to acidosis in crustaceans generally includes respiratory $CO_2$ excretion, $H^+$

excretion typically through $Na^+/H^+$ or $NH_4^+$ exchange and accumulation of extracellular $HCO_3^-$ as a buffer, where $HCO_3^-$ is derived through either branchial $Cl^-/HCO_3^-$ exchange and, to a lesser degree, from calcified structures (e.g. exoskeleton) (Wheatly and Henry, 1992). In freshwater crustaceans, acid-base regulation occurs mainly within the gills (Henry et al., 2012), where the $Na^+/K^+$-ATPase and $H^+$-ATPase generate the electrochemical gradients that drive ion exchange (Leone et al., 2017). The $Na^+/K^+$-ATPase alone may already account for more than 20% of an animal's energetic budget (Milligan and McBride,

1985), therefore, an increase in ion transport that must occur to re-establish and maintain acid-base homeostasis in the face of freshwater acidification could pose an increased energetic demand. In fact, in sea urchin larvae $pCO_2$ tensions of 800µatm have been shown to potentially double ion transport ATP demands (Pan et al., 2015). It is therefore conceivable that the energetic cost for long-term maintenance of acid-base homeostasis under freshwater acidification may come at substantial energetic cost which could have negative implications on other physiological parameters and thereby animal fitness.

Heightened energetic demands to maintain crucial physiological processes during exposure to environmental $CO_2$ acidification could be met through reallocation of energy budgets or through modification of metabolism to increase energy supplies. In fact, in marine brittle star *Amphiura filiformis* exposure to $CO_2$ tensions ranging from 1000-8000µatm for 40 days caused an increase in metabolic rate (increased energy budget) which was postulated to fuel increased calcification observed in this species (Wood et al., 2008). In contrast, the metabolic rate of juvenile European lobster (*Homarus Gammarus*) remained

unchanged when exposed to 1100 and 8000µatm $CO_2$; however, branchial $Na^+/K^+$ ATPase activity was increased demonstrating a reallocation of energy supplies despite maintaining an unchanged energy budget (Small et al., 2020). Unlike in juvenile European lobster and brittle star, juvenile Chinese mitten crab underwent a metabolic depression (decreased energy budget). Despite reductions in energy supplies crabs were still able to re-establish extracellular pH through $HCO_3^-$ accumulation suggesting a reallocation of energy supplies to essential ionoregulatory processes.





Typically, a metabolic depression as seen in present study is observed when an organism is unable to compensate for a reduction in extracellular pH (Pörtner et al., 2004). While in juvenile Chinese mitten crab this could be the case at the initial two days of the time course, by day seven extracellular pH was fully compensated yet metabolism remained depressed. It is known that high environmental $CO_2$ can trigger accumulation of compounds such as adenosine that can stimulate metabolic depression as occurs in the peanut worm, *Sipunculus nudus* (Reipschläger et al., 1997). A similar mechanism could conceivably

be in place that keeps Chinese mitten crab in a metabolically depressed state as a strategy to conserve energy supplies to promote survival upon exposure to short term stressors like high environmental $CO_2$. Such an adaptation may be present in Chinese mitten crab as these crabs would regularly experience short-term fluctuations in environmental $CO_2$ of their natural habitat. In fact, in the Mediterranean mussel (*Mytilus galloprovincialis*) a chronic metabolic depression lasting up to 90 days has been observed to allow survival following exposure to ocean acidification (5026 µatm $pCO_2$, Michaelidis et al., 2005).

While metabolic depression is a viable strategy used by many organisms to survive short-term periods of environmental stress (Guppy and Withers, 1999), it is a less viable long-term strategy as reduction in metabolic rate reduces energy availability for costly physiological processes such as calcification and protein synthesis which would ultimately affect growth and reproductive success as reported in freshwater pink salmon (*Oncorhynchus gorbuscha*) and marine amphipod (*Gammarus locusta*) (Borges et al., 2018; Ou et al., 2015).

In addition to overall metabolic depression, freshwater acidification led to an increase in extracellular concentrations and excretion of ammonia, a metabolic product of protein catabolism. Elevated excretion of ammonia may function as an excretable acid equivalent to assist the maintenance of pH homeostasis, a mechanism suggested for the brackish water green crab (*Carcinus maenas*) and hydrothermal vent crab (*Xenograpsus testudinatus*) (Allen et al., 2020; Fehsenfeld and Weihrauch, 2013). Furthermore, the previously mentioned observed reduction in oxygen consumption and increased ammonia excretion

(decrease in O:N ratio) indicates that juvenile Chinese mitten crab have a greater reliance on protein catabolism as an energy source under elevated environmental $CO_2$. Similar decreases in oxygen consumption and increases in ammonia excretion have been observed in the Mediterranean mussel (*M. galloprovincialis*, 5026µatm $pCO_2$, 15-90 days) and brittle star (*A. filiformis*, 6643µatm $pCO_2$, 28 days), where catabolism of amino acid such as glutamine may provide metabolic bicarbonate to further assist in sustaining pH homeostasis (Hu et al., 2014; Michaelidis et al., 2005). While potentially beneficial for sustaining acid-

base status, elevated protein catabolism requires a consistent source of protein through either a high protein diet or increased food consumption which if not met could result in muscle wastage an effect seen in brittle star during heightened energetic demands of ocean acidification (Wood et al., 2008). Further, the fact that feeding rate has been shown in juvenile European lobster (*H. gammarus*) and green crab (*C. maenas*) to decline as a result of elevated environmental $CO_2$ makes a greater reliance on protein catabolism during energetically constricted times a precarious situation for juvenile Chinese mitten crab

(Appelhans et al., 2012; Small et al., 2020).

Carapace calcification is an energetically costly process related to growth and predation defence in crustaceans that could be impaired by freshwater acidification and the associated metabolic changes. Generally, decapod crustaceans are believed to be the least susceptible of calcifying organisms to aquatic acidification as their exoskeletal $CaCO_3$ exists in the more stable calcite



form providing greater resilience to dissolution in contrast to bivalves and corals (Ries et al., 2009). Indeed, the marine
crustacean carapace is well protected from aquatic acidification mediated dissolution with reports of either no change or an
increase in calcification being typically observed (Kroeker et al., 2013; Ries et al., 2009; Whiteley, 2011). However, in the
present study, juvenile Chinese mitten crab had reduced levels of carapace calcification as reflected by a lower carapace
calcium content after three and six weeks of exposure (Fig. 4). While not as common, examples of reductions in carapace
calcification have been observed in marine crustaceans including several porcelain crabs and the tanner crab, *Chionoecetes*
*bairdi* (Long et al., 2013; Page et al., 2017). In crustaceans it has been suggested that carapace dissolution may occur to support
extracellular pH buffering that normally occurs through branchial $HCO_3^-$ uptake by providing an alternative source of $HCO_3^-$
(Cameron, 1985; Defur et al., 1980). In the present study, extracellular pH was recovered long before carapace dissolution was
apparent, therefore it is less likely that the carapace is mobilized as a source of $HCO_3^-$. Instead, reductions in carapace calcium
content most likely reflect an alteration in the rate of calcification or acid mediated dissolution of the carapace. As carapace
formation and maintenance is an energetically expensive process requiring careful ion regulation by numerous organs, the
aforementioned changes in whole animal energetics due to freshwater acidification could have negative implications on animal
fitness either by weakening the exoskeleton or impairing post-moult calcification which in turn can hamper growth and leave
animals vulnerable to predation.

**4.4 Behavioural Responses**

Elevated freshwater $pCO_2$ altered locomotory behaviour in juvenile Chinese mitten crabs. Crabs in acidified freshwater
covered less total distance during movement and did so at a lower velocity. No studies have previously examined changes in
crustacean distance covered in the presence of elevated environmental $CO_2$. However, reduced speed of movement has also
been report in Shiba shrimp (*Metapenaeus joyneri*) exposed to $CO_2$ levels of 9079μatm; however, unlike in Chinese mitten
crab this shrimp did not experience a reduction in resting/standard metabolic rate correlated with locomotory impairment
(Dissanayake and Ishimatsu, 2011). While not measured in our study, in Shiba shrimp there was a reduction in aerobic scope
which would likely lead to reduced aerobic performance and thereby reduced movement (Dissanayake and Ishimatsu, 2011).
Similar alterations in aerobic scope could partially be behind the reductions in velocity seen in juvenile Chinese mitten crab
however this is entirely speculative and there are many cases where elevated $CO_2$ does not alter aerobic scope (Lefevre, 2016).
In addition to moving slower, Chinese mitten crab spent less time moving their entire body throughout the novel arena and
less time moving only their appendages while staying at a fixed location. Reduced movement time and appendage movement
was also seen in the hermit crab (*Pagurus bernhardus*) exposed to 12000μatm $CO_2$ (de la Haye et al., 2011). In contrast, the
isopod (*Paradella dianae*) experienced no change in swim time or crawling time when exposed to 2085μatm $CO_2$ despite a
measured metabolic depression (Alenius and Munguia, 2012). Differences in the effect of $CO_2$ on movement time may be a
result of the $CO_2$ levels employed but further studies on a greater variety of species are required to determine potential patterns
for crustaceans. It is plausible that overall locomotory behaviour is reduced in this study due to alterations in neurological
function resulting from ionic imbalances or other $CO_2$-mediated effects that are known to occur from elevated environmental



$CO_2$ (For review of neural effects of aquatic acidification see Tresguerres and Hamilton, 2017). Additionally, with a reduction in overall energy availability, crabs may be reducing energy expenditure through locomotion to conserve energy stores for physiological processes more crucial to surviving the physiological distress caused by freshwater acidification. The overall

reductions in locomotion observed in juvenile Chinese mitten crab could have negative conseqeunces on their survival as reduced movement would make these crabs more vulnerable to predation, reduce migatory capabilites and reduce foraging ability.

## 5 Conclusion

In conclusion, we found impairment of survival, metabolism, calcification, and locomotion with exposure to a potential future

$CO_2$ mediated freshwater acidification scenario. Overall, energy availability was reduced despite heightened ionoregulatory energetic demands. Changes in the animals' energy budgets likely result in a greater dependency on protein catabolism as an energy source to allow for extracellular pH recovery at the cost of reducing their exoskeletal calcification and locomotion. We found that despite successful acid-base compensation, survivals rates declined with a 3.8 times greater probability of mortality under acidified freshwater conditions. While our study suggests negative impacts of freshwater acidification, these results

should be assessed with caution as the assumed acidification levels are based on a relationship between changes in atmospheric $CO_2$ and freshwater $CO_2$ which remains to be more effectively modelled. Nevertheless, this study shows that freshwater invertebrates may be at risk to future freshwater acidification and emphasizes the importance of modelling acidification in freshwater systems to accurately assess biological consequences of global change. Future biological studies should emphasize transgenerational adaptability, long-term effects of freshwater acidification on the scale of weeks to months, and assessment

of a wide range of freshwater species to determine animal performance indicators for $CO_2$ sensitive species as this will provide a far better understanding of the potential impacts of $CO_2$ mediated acidification in freshwater systems.

## Data Availability

Data are available at the following link http://dx.doi.org/10.6084/m9.figshare.13888034.

## Author Contributions

A.R.Q.R designed the study, performed experiments, analyzed data and wrote the manuscript. P-L.K, P-H.S, and M-T.H. performed experiments. G.J.P.A analyzed data and assisted with writing. P-P.H. provided financial support and analytical tools. Y-C.T. assited in designing the study, writing the manuscript, and provided financial support and analytical tools. D.W. assisted in designing the study, writing the manuscript, and provided financial support and analytical tools.

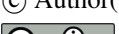



**Competing Interests**

The authors declare that they have no conflict of interest

**Acknowledgements**

Research was supported by the National Science and Engineering Research Counsel (NSERC DG; D.W.), the Ministry of Science and Technology, Taiwan, Republic of China (MOST 108=2621-M-001-003; Y.-C.T.), NSERC's PGS-D and University of Manitoba Graduate Fellowship (A.R.Q.R. and G.J.P.A.)

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
