# Peer review of "Anthropogenic CO2-mediated freshwater acidification limits survival, calcification, metabolism, and behaviour in stress-tolerant freshwater crustaceans"

_Biogeosciences, 2021_

## Referee Comment (RC3)

General. This paper reports results from experimental exposure of a freshwater crab to elevated $CO_2$ levels. The authors develop a justification based largely on the paucity of prior studies of biological $CO_2$ effects of freshwater taxa. The range of response variables is large and impressive including metabolic rates, a battery of physiological metrics, locomotor behavior, and survival. In total, the test organism was found to be CO2-sensitive in most of the responses measured which to actually runs counter to the authors' expectation for this taxon which happens to live in $CO_2$-variable habitats (more on this below). The paper provided a good summation of the literature and was reasonably well written (corrections and suggestions are identified below under 'Technical corrections').

Specific comments. I had two issues with the paper. The first is related to the packaging and broader context for the study. The second issue pertains to the analysis of the data.
Regarding the packaging of the study, the authors can make the paper clearer by lessening the effort to cover all bases as they describe the motivation and summary of their work. They begin this journey in the Introduction by providing context and underscoring the lack of studies on the effects of $CO_2$-induced acidification in freshwater taxa, especially calcifiers, relative to their marine counterparts. Fine, but one paper will not significantly change that balance. In fact, shortly after making this case, the authors add a 'but wait' because the Chinese mitten crab might not be sensitive to elevated $CO_2$ due to its occurrence in freshwater habitats that range widely in $CO_2$ concentrations. So it indeed might not be a representative freshwater taxon to study $CO_2$ effects after all. The reselling of mitten crab (actually, selling the paper about mitten crab) picks up again in the Discussion but from a different, all things to all people, vibe "we aimed to demonstrate for the first time the physiological and behavioural consequences of a possible future CO2 mediated freshwater acidification scenario on a juvenile calcifying invertebrate, the Chinese mitten crab." I would much prefer that the authors provide a more modest and clear appraisal of the import of their work and their working hypotheses, and let the readers assign value to their efforts.
The second issue is more problematic as it pertains to the experimental design (including response variables) and data analyses. The following points are interrelated and would benefit from a synthetic solution. One is suggested below.

1. It is unclear to me why the time-course data were run as a one-factor design (one-way ANOVA) rather than a two-factor design as was done for the other analyses. It would seem that a two-way analysis is appropriate and would provide the same hypothesis test as the one-way plus more (factor 1 vs factor 2 and test for the interaction between factors 1 and 2). Regarding the presentation of results in Table 1 from the series of one-way ANOVA's, the 'Treatment' appears to be incorrect as the control FW or the acidified FW treatment was only provided as one 'type' (i.e., the crabs were exposed to control FW in the 'Control FW' treatment OR were exposed to acidified FW in the 'Acidified FW' treatment, and not different levels of each factor) so what was being tested? Was it the change in response over time? If so, in this case 'Time' would be the 'Treatment' (aka factor) and Control FW or Acidified FW was the study condition. If I am interpreting the table correctly, the table needs to be restructured as does the language describing the tests.

2. Building on #1, it seems that many of the data types are based on repeated measures over time. Hence, a repeated measures or profile analysis would seem appropriate.

3. Building on #1 and #2, the response data collected in the experiments are highly interrelated. Within an experiment (e.g., locomotor behavior) the responses are clearly interdependent as well as repeated. Further, the authors make this very point in the Discussion, i.e., the responses are interrelated. For example, locomotion and energy expenditure in general are more labile in order possibly to conserve other responses more aligned with survival (Lines 403-04). There is likely a degree of nesting or hierarchy of responses (e.g., the locomotor behavior metrics are

more tightly interrelated responses within the entire set of responses used) but the authors DO argue in the Discussion that there is likely a network of inter-relatedness or covariance among the responses. If true (it likely is), then employing a set of univariate tests (i.e., the one-way and two-way univariate ANOVA's) is the wrong approach.

I would recommend revamping the statistical analyses. Run the data as two-way multivariate analyses (either MANOVA's and/or repeated measure designs). Discount the critical $p$-values to accommodate multiple tests on the same set of data (i.e., data on the same individuals in some cases or crabs drawn from the same tanks).

Technical corrections. Below are examples where corrections or reconsiderations are needed.

1. Be explicit when referencing the data, e.g., identify the response variables rather than refer to 'time-course data' (Line 160) as it would seem that multiple data types were recorded over time.
2. 'Data' is plural subject, e.g., data were.
3. Run spell check as there were misspellings (e.g., Line 405, 'conseqeunces')
4. Hyphens for compound adjectives (e.g., Line 129, 'closed-system respirometry') were commonly omitted
5. Inconsistencies in nomenclature after establishing a convention, e.g., SEM (Line 174) vs SE (Figure 1 legend).
6. Regarding the Conclusions, avoid closing with a wish list with little, if any, connection to prior text or the themes of this paper (e.g., Lines 418 – 420), "Future biological studies should emphasize transgenerational adaptability" – yes but seems out of place here. "... long-term effects of freshwater acidification on the scale of weeks to months" – allude to this theme earlier in ms. "...assessment of a wide range of freshwater species to determine animal performance indicators for $CO_2$ sensitive species" – this is the one of the three in this wish list that has a direct connection to this paper.

---

## Author Comment (AC1)

**Subject: Comment on bg-2021-34**

**Authors' responses are in italics**

Reviewer #3

Paper packaging: The first is related to the packaging and broader context for the study. The second issue pertains to the analysis of the data.
Regarding the packaging of the study, the authors can make the paper clearer by lessening the effort to cover all bases as they describe the motivation and summary of their work. They begin this journey in the Introduction by providing context and underscoring the lack of studies on the effects of $CO_2$-induced acidification in freshwater taxa, especially calcifiers, relative to their marine counterparts. Fine, but one paper will not significantly change that balance. In fact, shortly after making this case, the authors add a 'but wait' because the Chinese mitten crab might not be sensitive to elevated $CO_2$ due to its occurrence in freshwater habitats that range widely in $CO_2$ concentrations. So it indeed might not be a representative freshwater taxon to study $CO_2$ effects after all. The reselling of mitten crab (actually, selling the paper about mitten crab) picks up again in the Discussion but from a different, all things to all people, vibe "we aimed to demonstrate for the first time the physiological and behavioural consequences of a possible future CO2 mediated freshwater acidification scenario on a juvenile calcifying invertebrate, the Chinese mitten crab." I would much prefer that the authors provide a more modest and clear appraisal of the import of their work and their working hypotheses, and let the readers assign value to their efforts.

*We will modify the motivation and summary of our work to appear less like we are trying to sell this paper as a finite answer that freshwater calcifiers in general are at risk of freshwater acidification and focus more on the mitten crab. It was not our intent to state that all freshwater calcifiers are susceptible. We were more trying to get across the notion that this species is generally quite tolerant to environmental change yet is quite susceptible to future freshwater acidification and then use this to try and get the message across that this area requires greater focus so that the scientific community as a whole can make a more clear conclusion on the susceptibility of freshwater calcifiers to future freshwater acidification. We do acknowledge though that one study will not get the answer to this question but it definitely provides some important groundwork for the scientific community to build off.*

The second issue is more problematic as it pertains to the experimental design (including response variables) and data analyses. The following points are interrelated and would benefit from a synthetic solution. One is suggested below.

1. It is unclear to me why the time-course data were run as a one-factor design (one-way ANOVA) rather than a two-factor design as was done for the other analyses. It would seem that a two- way analysis is appropriate and would provide the same hypothesis test

as the one-way plus more (factor 1 vs factor 2 and test for the interaction between factors 1 and 2). Regarding the presentation of results in Table 1 from the series of one-way ANOVA's, the 'Treatment' appears to be incorrect as the control FW or the acidified FW treatment was only provided as one 'type' (i.e., the crabs were exposed to control FW in the 'Control FW' treatment OR were exposed to acidified FW in the 'Acidified FW' treatment, and not different levels of each factor) so what was being tested? Was it the change in response overtime? If so, in this case 'Time' would be the 'Treatment' (akafactor) and Control FW or Acidified FW was the study condition. If I am interpreting the table correctly, the table needs to be restructured as does the language describing the tests.

*We apologise for the confusion. The interpretation you have made was correct and it would have been better structured by placing the treatment as time and the control vs acidified as the study condition. We will make this correction.*

2. Building on #1, it seems that many of the data types are based on repeated measures over time. Hence, a repeated measures or profile analysis would seem appropriate.

*We will clarify this in the methods. This was not a repeated measures as we had 4 tanks with multiple crabs per tank and at each sampling time crabs were haphazardly selected from these 4 tanks. So, the population of crabs was repeatedly measured but as we could not guarantee that the same individuals were selected then we went with a non-repeated measures analysis.*

3. Building on #1 and #2, the response data collected in the experiments are highly interrelated. Within an experiment (e.g.,locomotor behavior) the responses are clearly interdependent as well as repeated. Further, the authors make this very point in the Discussion, i.e., the responses are interrelated. For example, locomotion and energy expenditure in general are more labile in order possibly to conserve other responses more aligned with survival (Lines403-04). There is likely a degree of nesting or hierarchy of responses (e.g.,the locomotor behavior metrics are more tightly interrelated responses within the entire set of responses used) but the authors DO argue in the Discussion that there is likely a network of inter-relatedness or covariance among the responses. If true (it likely is), then employing a set of univariate tests (i.e.,the one-way and two-way univariate ANOVA's) is the wrong approach.

I would recommend revamping the statistical analyses. Run the data as two-way multivariate analyses (either MANOVA's and/or repeated measure designs). Discount the critical p-values to accommodate multiple tests on the same set of data (i.e., data on the same individuals in some cases or crabs drawn from the same tanks).

*We would like to thank the reviewer for these in-depth comments about our statistical analysis. Upon reading the reviewer comments we agree that a revamp of the analysis by running a MANOVA would be more appropriate. As we did not use the same animals we would not use a repeated measure design as there was really no repeated measures.*

*Technical corrections suggested by reviewer 3 will be addressed*

---

## Author Comment (AC2)

**Subject: Comment on bg-2021-34**

**Authors' responses are in italics**

Reviewer #2

1.Whilst open ocean seawater is extremely consistent in its chemical composition (at least for a given salinity), freshwater is definitely not consistent, and in fact is extremely variable, in ways that can have major consequences for physiological responses to variables such as CO2/pH. It is therefore important to report details of the freshwater chemistry, more than just the carbonate chemistry in Table 1. In particular ion concentrations that are relevant for gill ion and acid-base regulation processes (e.g. sodium and chloride), and calcium is critical for understanding and interpreting potential calcification effects of the treatments.

*The ion composition of Taipei tapwater will be added into the methods section.*

2.To help the reader the authors should provide a conversion, or direct comparison, for pCO2 values reported in Pa and uatm.

*Thank you for pointing this out. We had meant to add this conversion and will do so in the revised version*

3.Some variables were measured over a 7 day exposure period (haemolymph acid-base, oxygen consumption rate and ammonia excretion rate), or 14 days (mortality), but others (calcification and behaviour) seem to be over 6 week exposure. These different timescales are not explained in Methods section, or justified.

*Clarification will be added into the methods section justifying the timescales for each experiment. For the calcification experiment it was briefly mentioned in the results (Line 224-225).*

4.What caused mortality? In particular could this have been related to cannibalism after individual crabs had moulted? This seems likely, as is common in crustaceans in aquaculture where animals have little chance to escape their conspecifics whilst waiting for their exoskeleton to harden after moulting. There is no mention of hides or shelters being provided in the exposure tanks, so if crabs were moulting it is possible that calcification was slower in the high CO2 treatment, resulting in more crabs being prone to cannibalism whilst waiting to calcify, rather than an inability to calcify eventually (given enough time). With 6-7 crabs in a 10 litre tank cannibalism seems likely if some were moulting.

*We did have some plastic pipe tubing in the tank for shelter (This will be clarified in the methods). In regard to what caused the mortality we cannot explicitly say as the crabs had no obvious signs of disease pointing to a reason for death. We can state that it was not due to cannibalism as we were always able to recover the intact bodies of the deceased crabs. From*

*our experience death due to cannibalism usually results in recovery of just parts of the crab and sometimes just shells of a recent moult.*

5.Why not calculate the actual ammonia quotient (AQ) and include discussion of these data regarding protein utilisation, and reference the AQ values found in other species and how these numbers relate to protein utilisation.

*We did not calculate the ammonia quotient or O:N ratio as the measurements of oxygen consumption and ammonia excretion were not done on the same animals at the exact same time. In the methods it is described how we did these two measurements. Since we don't have exact paired measurement of oxygen consumption and ammonia flux we cannot do an accurate calculation to provide a quantitative number with an accurate standard error. However, we do mention in the discussion (line 335) that O:N ratio appears to decrease as we have a reduction in O2 consumption but really no change in ammonia. Unfortunately, it was a methodological issue of being able to actual run the experiment long enough to detect ammonia without over depleting O2 that prevented us from measuring both simultaneously.*

6.In the locomotory behaviour tests (and metabolic rate and ammonia excretion rate measurements) it is important to report data for the carbonate chemistry variables actually measured (at the same time) in both the experimental holding tanks the crabs were taken from, and the arena tanks the behaviour was assessed in (or respirometers). This is important because if they were different $pCO_2$ values it could result in a rapid acid- base disturbance in the crabs transferred from one tank to another that could be the cause of behaviour differences or metabolic rate differences, rather than the actual prior high $pCO_2$ exposure.

*The water chemistry for the experiments would be the same as that reported for the tanks. We were running a flow through system so the build up of ammonia and other wastes was negligible. Therefore, we just collected water from the experimental tanks (this assured that total alkalinity was the same) and used a separate $CO_2$ controller and $CO_2$ tank to inject $CO_2$ into the container we were placing the water to be used for the experiment. Also, before using the collected water took a portable pH meter/probe and confirmed that pH in the experimental tanks and experimental water container were the same to assure that $pCO_2$ was the same. This detail can be added into the methods section, so the reader knows we have done due diligence to assure the water parameters for the experiment were maintained as identical to the experimental tanks as possible.*

7.There is considerable discussion of the data showing a metabolic depression caused by freshwater acidification. However, if I understand the Methods accurately, metabolic rate (as oxygen consumption rate) was only measured for a single 30 minute period in each crab, and this was only after 15 minutes "acclimation" following handling and transfer to the respirometer chamber. If this is the case, then what was measured cannot be considered as the stable metabolic rate during exposure to either treatment (low or high CO2), and "metabolic depression" is not an accurate conclusion to make. Instead, what was measured is more likely to be the acute metabolic response to handling, brief air- exposure and transfer to a new environment, on top of the effects of any prior exposure to the CO2 levels used. This has not been considered but is important in interpreting the data reported.

*Your interpretation of the methods is correct. We do acknowledge that this is not the perfect way of doing measurements of oxygen consumption but is a widely published approach used for crustaceans. From our experience on other crustaceans using intermittent flow respirometry crustaceans do not typically require super long rest times for metabolic rate to stabilize. However, as we cannot explicitly say that is the case for our study, we will mention in the manuscript that the handling stress and brief air exposure are caveats for the readers to consider when interpreting the results and conclusions we make. We are all for transparency in our research and believe that there is still value in these results. We also believe this does show a metabolic depression as both animals were treated the same but with the caveat that the handling stress and brief air exposure must be considered when assessing what our results indicate.*

8.The manuscript often refers to "calcification" being measured, but this implies the rate of calcification which was not actually measured. Instead, carapace calcium content was measured at a few timepoints, which has been used to imply "calcification rate", but that is not strictly true. See also comments above about moulting, immediately after which is when the greatest rates of calcification occur.

*We apologise for the confusion. This issue basically comes down to how one interprets calcification. By definition it is simply the build up of calcium salts on a tissue and not necessarily a rate. As we have measured the calcium content in the carapace we have indeed measured calcification but not a calcification rate. I have double checked the manuscript and can confirm we never state that we are measuring a calcification rate. In the methods line 140 we state that we are assessing carapace calcium content as a proxy of calcification. Based on this information we do not believe any of our statements about calcification measurement is false.*

9.L.77-78 – The description of how pH/CO2 was controlled is not sufficiently detailed to provide a full explanation. Presumably this was done using 4 pH electrodes permanently recording the pH in each of the 4 individual experimental tanks, and the signals received from each electrode by 4 separate pH controllers was used to regulate the flow of CO2 via air stones into these individual tanks? Please provide enough details to clarify this issue.

*Additional details will be provided in text. Essentially your description is accurate and there were multiple CO2 controllers each with their own pH probe and mini CO2 tank that was regulating a single 10L aquaria.*

10. l.86 – Given that the CO2sys program requires salinity as an input variable to calculate carbonate chemistry, how was salinity measured, and what value(s) were used in these calculations?

*The CO2sys program has a freshwater function where they essentially count salinity as 0. This function was used for the calculations.*

11. Table 1 – It is not clear what these data are reporting, i.e. what timepoints do these data represent? From which experiments (the 7 day, 14 day or 6 week experiments?), and how were

the means calculated with respect to the four different replicate tanks per treatment? More details are needed. It would seem appropriate to report data separately for the different duration experiments (7 day, 14 day or 6 week).

*Clarification of this will be provided in the revised manuscript.*

12. l.115 – How were the crabs selected "randomly"? Unless a truly random method was used, this usually means the first animals that were able to be caught be experiments, which can result in a bias based on behavioural traits of the animals.

*We will change the wording as this was more of a haphazardly selecting crabs from the four tanks.*

13. l.119 – All units should be separated from their number by a space. So the 200mL should be 200 mL. This comment also applies throughout the whole manuscript.

*We will check for this and make the change throughout*

14. The control values for haemolymph pH are very high (pH >8.1) for an aquatic animal at the temperature used (23 degrees C). The haemolymph bicarbonate is also surprisingly high (13-14 mM) in the control conditions (time zero for both treatments). Studies on other crustaceans suggest haemolymph pH at this temperature would be closer to 7.6-7.8 and bicarbonate closer to 3-6 mM, and usually only reach values this high if the animals were already exposed to very high $CO_2$ (e.g. >10,000 uatm) and had accumulated bicarbonate to compensate pH. Perhaps there is a precedent for such high bicarbonate and pH in this species, or other crab species, that I am not aware of, but the authors provide no discussion or comment on this discrepancy.

*This might be something specific to this species as it is quite a high pH and bicarbonate level compared to other crustaceans we have come across although those are mostly marine. Nevertheless, the values we measured in our study are comparable to what has been previously measured in adult E. sinensis (See Truchot 1992 Resp Physiol 87 419-427). A mention of this abnormal levels can be added into the text if deemed necessary although this is not the central focus point of the study.*

---

## Author Comment (AC3)

**Subject: Comment on bg-2021-34**

**Authors' responses are in italics**

**Reviewer #1**

1. **Length of experiment**- In the methods section, the authors need to state the duration of their experiments. I noticed on the results table that for some measurements, the time axis extended to 8 weeks while in others 7 days, so I'm assuming that different experiments were done for different lengths of time? If so this needs to be explicitly stated somewhere in the methods.

*In the methods section we will clearly state the timeline of exposure for each measured parameter.*

2. **Line 37: "To date there are no comprehensive studies investigating various physiological and behavioral effects of realistic future levels of CO2- mediated acidification in calcifying freshwater invertebrates".** This is false. David et al. (2020) investigated the effects of CO2-induced acidification on the invasive freshwater gastropod, *Viviparus georgianus* over a 12 week period (Journal of Molluscan Studies 86:259-262). They used shell repair as a proxy for physiological performance. The findings of that study was the first to assess the effects of CO2-induced acidification in a calcifying freshwater invertebrate and regardless of whether the authors think it was 'comprehensive' or not, that reference should still be somewhere in the manuscript and the results compared with theirs.

*We respectfully disagree with the reviewer on the inclusion of David et al. 2020 study within our manuscript. We would like to point out that we stand by the statement that there are no **comprehensive** studies that use **realistic future** $CO_2$ levels. While other studies have investigated elevated $CO_2$ in freshwater animals the $CO_2$ levels employed are typically far beyond what could realistically be expected for near future FW acidification. In the David et al. 2020 study mentioned by the reviewer the effects of $CO_2$ induced acidification on shell growth rates of a freshwater gastropod were measured. In that study, the acidification treatments were based on a $CO_2$ induced pH drop from pH 7.3 to pH 6.8 and 6.3. Unfortunately, this manuscript does not provide key water parameter data (water alkalinity, water total carbon or water $pCO_2$ levels) that allows any reader to accurately quantify whether the treatments used are within a range that could realistically be seen in future FW systems. In addition, since the treatment $pCO_2$ levels are unknown and sufficient data has not been provided for the reader to make the calculation themselves it is difficult to then take this study and make a direct comparison to our study as it is crucial in making comparisons between freshwater/ocean acidification studies to know the $CO_2$ levels employed.*

3. What was the rationale for using the Chinese mitten crab for this study? No information is provided on the study species in terms of life history, fecundity, etc? The authors mentioned that it is a model organism for studying climate change but do not actually explain why.

*We will add more information on the reasoning for using Chinese mitten crab in the manuscript. We do not state it is a model organism for studying climate change but state that it could be an interesting model as this species is an invasive species that is quite tolerant to a range of environmental parameters.*

4. Very little is mentioned on the invasive status of Chinese mitten crabs in other parts of the world and nothing is mentioned about the implications of the findings of this study on the management/control of invasive populations of the species. This is what I meant when I mentioned earlier that the authors did not put their findings into a broader context.

*As mentioned above we will add a bit more about our reasoning for using Chinese mitten crab and their invasive status/importance in aquaculture in Asia. In terms of trying to extend our results into the implications of this study on management and control of invasive populations we would more likely want to refrain from making too bold of claims as this is really a first probing study into the effects of freshwater acidification and our results here do not necessarily translate into how future populations will actually be affected as we are not considering multiple factors such as generational adaptation to changing environments. We would like to try and avoid over-reaching the implications of this study into say management and control of invasive species as we believe an initial probing study shouldn't be trying to make that bold of a claim but more provide the grounds for further studies that can build on our findings.*

5. **Line 65:** The authors mentioned that these crabs were 'purchased'; does that mean that they were bred in a controlled setting prior to arrival? If so, how sure are you that their physiological responses are reflective of what happens in natural populations?

*Yes, the crabs were purchased. However, our source wild catches their crabs and then rears them to adulthood. Since we were purchasing juvenile crabs they would have been early wild caught crabs and were then maintained under lab conditions for several weeks prior to experiments. This will be clarified in the manuscript as they are not traditionally aquaculture bred and reared crabs but derived from wild stocks.*

6. **Line 71:** Is oatmeal and mollusc meat the type of food the crabs would usually eat in their natural environment? Is that is a best practice for doing controlled experiments on crabs in the lab? (If so please provide reference).

*Chinese mitten crab are opportunistic omnivores that in early life mainly eat plant material but become more carnivorous as they grow. Studies have essentially shown they eat plant material, small invertebrates (including bivalves), and injured/dying/dead fish. We chose oatmeal due to personal communication for our aquaculture source that this is something they do feed the crabs. We will add a citation of a report from the department of fisheries and oceans Canada supporting this dietary behaviour.*

7. If you collected juveniles and cultured them under these stressors, why not measure the growth rate? In fact, why wasn't size measurements taken prior to each experiment?

*Unfortunately, due to logistic reasons we were not able to fit a growth experiment into this study. The vast majority of our experiments were not done over an extended period of time and as crustaceans must molt to grow the length of our experiment would unlikely allow for an accurate analysis of growth rates. We did perform the carapace calcium content experiment over 6 weeks but as we were periodically sampling, we would have only had a sample size of roughly 8 crabs that made it to the 6 week point which we feel would be too small of a sample size for proper growth experiments.*

8. What was the sample size for each experiment. This needs to be included either in the methods or when reporting statistics in the results section.

*Sample size for each experiment is present in the figure captions but can be written into the text if necessary.*

---

## Author Response (AR1)

**Subject: Comment on bg-2021-34**
**General Comments:**
We would like to thank the three reviewers as the in-depth critical comments have vastly improved the analysis and overall quality of this manuscript. Detailed response to all concerns is provided below.

**Reviewer #1**

1. Length of each experiment has been included in the methods section

2. We respectfully disagree with the reviewer on the inclusion of David et al. 2020 study within our manuscript. We would like to point out that we stand by the statement that there are no **comprehensive** studies that use **realistic future** $CO_2$ levels. While other studies have investigated elevated $CO_2$ in freshwater animals the $CO_2$ levels employed are typically far beyond what could realistically be expected for near future FW acidification. In the David et al. 2020 study mentioned by the reviewer the effects of $CO_2$ induced acidification on shell growth rates of a freshwater gastropod were measured. In that study, the acidification treatments were based on a $CO_2$ induced pH drop from pH 7.3 to pH 6.8 and 6.3. Unfortunately, this manuscript does not provide key water parameter data (water alkalinity, water total carbon or water $pCO_2$ levels) that allows any reader to accurately quantify whether the treatments used are within a range that could realistically be seen in future FW systems. In addition, since the treatment pCO2 levels are unknown and sufficient data has not been provided for the reader to make the calculation themselves it is difficult to then take this study and make a direct comparison to our study as it is crucial in making comparisons between freshwater/ocean acidification studies to know the $CO_2$ levels employed.

3. We have restructured the final paragraph of the introduction to clarify why the Chinese mitten crab was selected as our model freshwater crustacean.

4. As mentioned above we have restructured the introduction to include our reasoning for using Chinese mitten crab and their invasive status/importance in aquaculture in Asia. In terms of trying to extend our results into the implications of this study on management and control of invasive populations we would more likely want to refrain from making too bold of claims as this is really a first probing study into the effects of freshwater acidification and our results here do not necessarily translate into how future populations will actually be affected as we are not considering multiple factors such as generational adaptation to changing environments.

5. We have amended the methods to let the reader know that these are purchased wild caught crabs.

6. Chinese mitten crab are opportunistic omnivores that in early life mainly eat plant material but become more carnivorous as they grow. Studies have essentially shown they eat plant material, small invertebrates (including bivalves), injured/dying/dead fish. We chose oatmeal due to personal communication for our aquaculture source that this is something they do feed. We have added a citation supporting their dietary behaviour.

7. Unfortunately, due to logistic reasons we were not able to fit a growth experiment into this study. Most of our experiments were not done over an extended period and as crustaceans must molt to grow the length of our experiment would unlikely allow for an accurate analysis of growth rates. We did perform the carapace calcium content experiment over 6 weeks but as we were periodically sampling, we would have only had a sample size of roughly 8 crabs that made it to the 6 week point which we feel would be too small of a sample size for proper growth experiments.

8. Sample size for each experiment is present in the figure or table captions.

**Reviewer #2**

1. The ion composition of Taipei tap water has been added into the methods section as measured by chloride assay and atomic absorption spectrophotometer.

2. The conversion between Pascals and micro atmospheres has been provided in the figure caption where hemolymph acid-base status is presented and in text of the results section where hemolymph acid-base status is first presented.

3. Clarification of the timescale for each experiment was added into the methods section.

4. We have added in our methods section that PVC pipes were present for shelter in all tanks. In regard to what caused the mortality we cannot explicitly say as the crabs had no obvious signs of disease pointing to a reason for death. We can state that it was not due to cannibalism as we were always able to recover the intact bodies of the deceased crabs. From our experience death due to cannibalism usually results in recovery of just parts of the crab and sometimes just shells of a recent moult. We have added mention in the discussion section that the mortalities in our study were unlikely due to disease or cannibalism as there were no obvious signs pointing to these explanations. That being said we can never 100% rule out disease as a reason.

5. We did not calculate the ammonia quotient or O:N ratio as the measurements of oxygen consumption and ammonia excretion were not done on the same animals at the exact same time. In the methods it is described how we did these two measurements. Since we don't have exact paired measurement of oxygen consumption and ammonia flux we cannot do an accurate calculation to provide a quantitative number with an accurate standard error. However, we do mention in the discussion (line XXX) that O:N ratio appears to decrease as we have a reduction in O2 consumption but really no change in ammonia. Unfortunately, it was a methodological issue of being able to actual run the experiment long enough to detect ammonia without over depleting O2 that prevented us from measuring both simultaneously.

6. The water chemistry for the experiments would be the same as that reported for the tanks. We were taking water from the same source that was feeding our experimental tanks and then acidifying them with a pH controller. Using a portable pH probe we then measured the water being used for the experiment to the tanks to make sure we were maintaining the exact same $CO_2$ tension.

7. Your interpretation of the methods is correct. We do acknowledge that this is not the perfect way of doing measurements of oxygen consumption and we would have benefited from an intermittent flow respirometry approach. However, due to our time course approach and the number of animals we had to process in that time this was the most experimentally feasible approach. We believe the best course of action in this case is to add a comment in the methods acknowledging the methodological limitations of our approach and refraining from calling our result a metabolic depression but instead refer to it as a reduced oxygen consumption rate. While our close system respirometry approach is not ideal we would like to note that we have preliminary data on lobster, crayfish, and green crabs using an intermittent flow approach showing that at least in these crustaceans oxygen consumption rates level off after 30 minutes in the experimental chamber. These experiments on crayfish, lobster, and crab were done by transferring animals from their holding tank into respirometry chamber and recorded over a 24-hour period.

8. We apologise for the confusion. This issue basically comes down to how one interprets calcification. By definition it is simply the build up of calcium salts on a tissue and not necessarily a rate. As we have measured the calcium content in the carapace we have indeed measured calcification but not a calcification rate. I have double checked the manuscript and can confirm we never state that we are measuring a calcification rate. In the methods line 140 we state that we are assessing carapace calcium content as a proxy of calcification. Based on this information we do not believe any of our statements about calcification measurement is false.

9. Additional details have been provided in text regarding our aquarium $CO_2$ regulating setup. Essentially there were multiple CO2 controllers each with their own pH probe and mini $CO_2$ tank that was regulating a single 10L aquaria.

10. The CO2SYS program has a freshwater function where salinity is counted as 0. This function was used for the calculations and does not require a measurement of salinity.

11. Thank you for pointing out this mistake. Our table should have included the water parameters for the 7-day, 14-day and 42-day (6 week) experiment. It was our mistake that we had only added the data for the 7-day experiments but have now amended the table to include the water parameters for 14-day and 6-week experiments. We also made sure to recalculate all the water parameters in CO2SYS to assure accuracy in the reported values and the new table values have been confirmed to be accurate. It should also be noted that the previously reported tank temperature was mean +/- SD but we have now changed to +/- SEM.

12. We have changed the wording from randomly selecting crabs to haphazardly as there was no set randomization to the selection but crabs were just selected alternating between tanks.

13. Spacing between numbers and units have been changed throughout the text.

14. The hemolymph pH and $HCO_3^-$ levels measured in this study are on the higher end of that normally seen in crustaceans which we would place around pH 7.6-7.9 and 3-9mM. That being said there are many reported cases of pH in the range we have detected (see book chapter Fehsenfeld and Weihrauch 2017). Nevertheless, the values we measured in our study are

comparable to what has been previously measured in adult E. sinensis (See Truchot 1992 Resp Physiol 87 419-427). As these values are not unusual for this species we haven't really gone into that as this is not really the central topic of this study.

**Reviewer #3**

1. We have restructured the paper so that it focuses more on what our results mean for Chinese mitten crab and avoid trying to overstretch our conclusions to try and encompass freshwater crustaceans in general
2. We appreciate your detailed review of our statistical analysis and have taken into consideration your concerns. We did not run the analysis as repeated measures because it was not a true repeated measure. The 4 different tanks were repeatedly measured but we cannot be certain that the same individuals were selected at each sampling point so avoided a repeated measures approach. Regarding the use of MANOVAs, we consulted with a colleague more familiar with this type of analysis. We found that our data violates the assumption of absence of co-linearity among dependent variables and the multiple normality test did not pass despite each dependent variable being normally distributed. For these reasons the MANOVA approach was avoided. In the end we took your advice to use a two-way ANOVA approach with a post hoc Dunnett's test as we are aiming to compare to how each group of crabs changes relative to the zero-day time point.
3. Suggested technical corrections have been addressed.

---

## Author Response (AR2)

**General comments:**
**We would like to thank the editor for the constructive comments that have vastly improved overall quality of this manuscript. All suggested comments have been addressed in text.**

Your revised manuscript has been reviewed by myself and I determined that you have sufficiently addressed the comments of the reviewers. However, there are a few editorial changes needed that would be easier done now than at the proofing stage. They are:

(1) Please review the text that you have added. Some of the sentences need to be further clarified and the grammar checked.

**We have gone through the text and altered sentences to improve clarity and grammar.**

(2) Table 2 caption should better relate to the columns in the table. For example, there should be "dependent and independent variables" stated in the caption.

**Table 1 and 2 captions have been altered**.

(3) While you point out the missing information in the David et al. (2020) paper and therefore refuse to cite it, you should still cite it but in the Introduction or Discussion. It is a recent paper that is relevant to your study. You are not limited to citing it only if you can directly compare it to your methods and results - cite it in a more general way.

**Line 39-40 We have added a sentence acknowledging that while our study is the first investigating relevant future $CO_2$ freshwater acidification that other studies like that of David et al. 2020 have used elevated $CO_2$ to investigate effects on physiology of calcifying invertebrates but at levels far higher than what could be expected in the future.**